# Resilin Distribution and Abundance in *Apis mellifera* across Biological Age Classes and Castes

**DOI:** 10.3390/insects14090764

**Published:** 2023-09-13

**Authors:** Audrey Anderson, Noah Keime, Chandler Fong, Andrew Kraemer, Carol Fassbinder-Orth

**Affiliations:** 1Department of Biological Systems Engineering, University of Nebraska-Lincoln, 1400 R Street, Lincoln, NE 68588, USA; aanderson111@huskers.unl.edu; 2Biology Department, Creighton University, 2500 California Plaza, Omaha, NE 68178, USA; 3Independent Scholar, Omaha, NE 68127, USA; andrew.c.kraemer@gmail.com

**Keywords:** honey bee, resilin, polyethism, wings, ddPCR

## Abstract

**Simple Summary:**

Insects possess a very strong and flexible protein in their appendages and wing joints called resilin. In flying insects, resilin provides wing flexibility that improves flight and movement dynamics. In our study, we investigated how resilin changes with age in honey bees. We discovered that resilin genes are expressed at high levels in worker honey bee pupae, but then levels decline significantly with age. We also determined that the resilin protein is found at higher levels in some joints in the newly emerged honey bees compared to older age classes, further confirming the age-dependent nature of resilin presence in honey bee wings. Specifically, resilin is highly expressed in pupae, deposited at high amounts in some joints in newly emerged honey bees, and then expression and deposition both decline rapidly with age. Resilin must be kept hydrated to stay functional, so these results suggest that although resilin genes may not be expressed significantly after emerging, the resilin protein is likely maintained through fluid support to allow for strong flight dynamics throughout life. Resilin expression and maintenance should be explored for their potential to be used as an indicator of wing development and overall wing health in aging honey bees in the future.

**Abstract:**

The presence of resilin, an elastomeric protein, in insect vein joints provides the flexible, passive deformations that are crucial to flapping flight. This study investigated the resilin gene expression and autofluorescence dynamics among *Apis mellifera* (honey bee) worker age classes and drone honey bees. Resilin gene expression was determined via ddPCR on whole honey bees and resilin autofluorescence was measured in the 1m-cu, 2m-cu, Cu-V, and Cu2-V joints on the forewing and the Cu-V joint of the hindwing. Resilin gene expression varied significantly with age, with resilin activity being highest in the pupae. Autofluorescence of the 1m-cu and the Cu-V joints on the ventral forewing and the Cu-V joint on the ventral hindwing varied significantly between age classes on the left and right sides of the wing, with the newly emerged honey bees having the highest level of resilin autofluorescence compared to all other groups. The results of this study suggest that resilin gene expression and deposition on the wing is age-dependent and may inform us more about the physiology of aging in honey bees.

## 1. Introduction

*Apis mellifera*, the European honey bee, is the single most important species of pollinators to human agriculture. Its wide distribution and use of generalist strategies allow *A. mellifera* to be of importance to global natural habitats and agricultural production [1]. As with other eusocial insects, honey bees exhibit a division of labor within their colonies. The division of labor in *A. mellifera* colonies during the brood-rearing months is a temporary physical caste system, where work is based on age and morphological specialization [2,3,4]. This labor strategy maximizes efficiency and allows for plasticity between adult worker biological age classes: newly emerged, nurse, guard, and forager [4,5,6]. Temporal polyethism in *A. mellifera* follows a “push-pull” model in which the nurse class is “pushed” into the guard class by the development of the newly emerged honey bees and the guard honey bees are “pulled” from classes by the population of foragers [4]. Flexible maturation, dependent upon colony needs, increases resiliency against environmental changes. When a biological age class loses a significant number of members, workers from other age classes will reverse or accelerate maturation to maintain the crucial nurse:guard:forager ratio [7].

The forager class flies the farthest distance of any biological age class, making wings and flight ability crucial to their role. *A. mellifera* has many advantageous flight adaptations—indirect flight mechanisms and asynchronous flight allow for increased flapping flight frequencies; rows of hamuli link the forewing to the hindwing, creating a larger airfoil surface that generates greater lift; and the presence of wing joints embedded with resilin [8,9,10,11]. Insect wings consist of a thin cuticle membrane, supported by chitin longitudinal and cross veins. The intrinsic musculature of an insect is restricted to the wing base, not extending into the wing, thus making them reliant on passive, flexible deformations of the wing during flapping flight [8,11,12,13,14]. Wing flexibility is beneficial to *A. mellifera* flapping flight. It increases load-lifting capacity, which the foragers need to carry pollen; increases vertical force production; reduces drag ratios; increases body stability and decreases bodily rotation during turbulent airflow; increases chamber and cupping, which aid in lift; reduces energy expenditures by nearly 25%; and passively stabilizes the insect while hovering [13,14,15,16,17]. This flexibility is dependent upon resilin, an elastoprotein found in insect cuticle.

Insect wings are composed of two layers of integument. Each of these layers contains cuticle, a chitinous, extracellular matrix. The wing veins contain six layers of cuticle. Inside the vein, hemolymph, neurons, tissues, and tracheae exist, hydrating the cuticle of the wing membrane and allowing the membrane to retain its elasticity [18,19,20]. Throughout the body of the insect, hypodermal cells inside the living integument secrete endocuticle materials daily. As new layers are added, resilin is deposited by a process known as endocuticular deposition.

Resilin is the most efficient elastomer known with 97% resilience under high-frequency conditions and a fatigue lifetime of 300 million cycles [21,22,23]. The two amino acids that comprise resilin, dityrosine and trityrosine, autofluorescence in neutral and alkaline conditions at excitation maxima of 318 and 323 nm, respectively, and both at an emission maximum of 420 nm [24,25]. Resilin is a structural protein found throughout insect cuticle, and its presence in wing vein joints allows for wing flexibility during flapping flight [26,27,28]. The resilin patches increase chordwise flexibility, reduce stress, improve efficiency, store elastic energy, and aid in the return of the wings to their original position after elastic deformation [8,15,17,26,27,29,30]. Resilin is distributed along flexion lines, the flapping flight axes of deformation [26,27].

Previous studies on the dragonfly *Epiophlebia superstes* have indicated that resilin distribution on wings varies between species and individuals of the same species [9,31,32]. Other studies have found that *A. mellifera* workers experience physiological variations between biological age classes including, but not limited to, expression levels of vitellogenin and adipokinetic hormone receptors, stress resistance, and fat body sizes [33,34,35,36]. Resilin distribution has been mapped on guard honey bees at the first medio-cubital (1m-cu), second medio-cubital (2m-cu), cubito-vannal (Cu-V), second cubito-vannal (Cu2-V), first radio-medial (1rs-m) cross vein and along the first and second radial sector of the forewing, and at the cubito-vannal (Cu-V) vein joint of the hindwing [27]. However, the potential of resilin to vary among *A. mellifera* biological age classes and be an age-dependent predictive marker of wing development has not been reported. This study explored the hypothesis that resilin varies among biological age classes in honey bees.

## 2. Materials and Methods

### 2.1. Specimen Collection

Honey bees were collected from Bountiful Blossoms Bee Company, LLC in Glenwood, IA, USA from May–June 2019. Biological age class was estimated on-site using class-specific characteristics and behavioral norms [37,38]. Specifically, purple-eyed pupae were collected for the pupae class, representing a distinct age class of approximately 7–10 days post-oviposition, workers actively emerging out of cells as the newly emerged class, workers actively tending to the nest as nest honey bees, and workers returning to the hive with pollen on their legs as foragers. Drones at the entrance of the hive were collected for drone specimens. Honey bees were collected from 5 different hives for each class. For gene expression analysis, honey bees were immediately placed in RNAlater and then stored at −80 °C until further use. For autofluorescent analysis of resilin on wings, honey bees were collected in wooden cages and then frozen at −20 °C until further use.

### 2.2. Resilin Autofluorescence 

#### 2.2.1. Dissection and Slide Preparation

For each honey bee, the wings were dissected along the axillary sclerites. Each wing was first swabbed with 70% ethanol and then secondly with a solution of 50% Triton X and 50% distilled water to clean off pollen and other particles on the wing. The fluorescence of resilin is sensitive to pH: in neutral and alkaline media, the excitation/excitation maxima are 320/420 nm, while in an acidic media, they are 285/420 nm [24]. The wash protocol of ethanol, Triton X-100, and distilled water resulted in wings being exposed to a primarily neutral solution (pH 7.2–7.3), which was taken into account when selecting the best filter for autofluorescence detection. After being swabbed, the wings were covered and dried. The microscope slides were cleaned using 70% ethanol and wiped to prevent dust contamination. The wings were arranged, polyurethane adhesive was swabbed along the edges of the microscope slide, and another microscope slide was placed on top. Replacing the coverslip with a second microscope slide allowed the increased weight to flatten the natural camber of the wing, reducing the number of focal planes during imaging. Procedures were derived from previous techniques and adapted to this study [27,39].

#### 2.2.2. Microscopy

Wings were observed using fluorescence microscopy on an Olympus BX61 microscope (Olympus Scientific Solutions Americas Corp., Waltham, MA, USA) equipped with an aniline blue filter (excitation maxima: 377 nm; emission maxima: 409 nm). Images were taken with an Olympus SC100 Color Camera that was connected to the microscope. Initial images of resilin were taken with samples and underwent tests to determine the exposure that allowed for the greatest clarity of resilin across multiple joints and focal planes without saturation. All images were taken at 20× magnification and 450 ms exposure. The dorsal, ventral, left, and right sides of all wings were imaged at the first medio-cubital (1m-cu), second medio-cubital (2m-cu), cubito-vannal (Cu-V), and second cubito-vannal (Cu2-V) joints on the forewing and the cubito-vannal (Cu-V) joint on the hindwing (Figure 1).

#### 2.2.3. Image Analysis

All images were blinded by a third party and uploaded to ImageJ version 15.2 (U. S. National Institutes of Health, Bethesda, MD, USA [40]). The scale was set and the following measurements were selected: area, minimum and maximum gray value, mean gray value, and integrated density. The picture identification code, honey bee and hive identification code, and orientation were recorded, and scale bars were burned into each image.

The autofluorescent resilin was manually traced three times and the average for each measurement was found. Then three selections of the background were taken and the average for each measurement was found. The corrected total cell fluorescence (CTCF) equation, a quantitative fluorescence imaging analysis, was used to calculate the final measurement: CTCF = Mean Integrated Density − (Mean area of the cell * Mean Gray Value of the background) [41]. CTCF can be used for any fluorescent material within a restricted area. CTCF calculates the fluorescence through the integrated density measurement, which finds the summation of gray pixel values within a selection and multiplies it by the area of the selection. Average CTCF was calculated because it incorporates the area and intensity of resilin fluorescence into one standard number while simultaneously subtracting out background fluorescence. Average CTCF was measured in relative fluorescence units (RFU). RFU is a standard measure of fluorescence intensity.

### 2.3. Resilin Gene Expression

#### 2.3.1. RNA Extraction

To prepare RNA for downstream reverse transcription and droplet digital PCR (ddPCR), the RNA from whole honey bees stored in RNAlater was extracted using the Qiagen RNeasy Mini Kit (Qiagen Inc., Valencia, CA, USA) with some adjustments as per the manufacturer’s instructions. Initially, the samples were homogenized in RNAlater using cold, sterile zirconia beads and a bead beater. A small portion of the homogenate was then mixed with lysis buffer (buffer RLT with dithiothreitol (DTT)). After another round of homogenization in the lysis buffer and subsequent centrifugation, the resulting supernatant was transferred to a separate tube containing 70% ethanol to eliminate any debris that could potentially obstruct or harm the Qiagen RNeasy spin column membrane. The remaining steps for RNA isolation followed the manufacturer’s protocol.

#### 2.3.2. Reverse Transcription

For reverse transcription of samples, the Bio-Rad iScript cDNA Synthesis kit (Bio-Rad Laboratories, Hercules, CA, USA) was employed following the manufacturer’s instructions. The following thermocycler settings were utilized: priming for 5 min at 25 °C, reverse transcription for 20 min at 46 °C, and reverse transcription inactivation for 1 min at 95 °C. The samples were subsequently stored at −80 °C until they were required for further analysis.

#### 2.3.3. ddPCR

Droplet digital PCR was completed according to the manufacturer’s protocol using the Bio-Rad QX200 ddPCR System. Pro-resilin gene expression analysis was used to make inferences about resilin deposition timing as it related to worker age classes. Pro-resilin is the precursor protein that gets transported to the subcuticular space before crosslinking to form a functional resilin protein network [42]. The concentration of cDNA was normalized to a housekeeping gene, eukaryotic translation initiation factor 3 subunit C (eIF3-S8). eIF3-S8 was chosen due to its use in previous qRT-PCR gene expression studies as the reference gene for *A. mellifera* and its similar magnitudes of expression compared to pro-resilin [43,44,45].

A ddPCR was then performed according to Tokach et al. [46]. Briefly, primers and probes were utilized at concentrations of 300 nM and 150 nM, respectively (Table 1). A PCR was performed using the C1000 Touch thermocycler (Bio-Rad Laboratories). The thermocycler was programmed with the following settings: (1) 10 min at 95 °C, (2) 30 s at 94 °C, (3) 1 min at 57.5 °C, (4) repeat steps 2 and 3 for a total of 39 cycles, (5) 10 min at 98 °C, and (6) 20 min at 4 °C. Additionally, a ramp rate of 2.5 °C was applied to Steps 2–4. Samples were then transferred to the Bio-Rad QX200 Droplet Reader. The output from the droplet reader was analyzed using Quantasoft software version 1.7 (Bio-Rad Laboratories, Hercules, CA, USA)to determine the copy number of target sequences.

### 2.4. Statistical Analyses

#### 2.4.1. Resilin Gene Expression by Age Class

As there was unequal variance in resilin expression among age classes (F_4,161_ = 18.29; *p* < 0.001), we used the ‘lm.rrpp’ function from the ‘RRPP’ R package to perform a permutational ANOVA, using age class as the explanatory variable and resilin expression as the response data.

#### 2.4.2. Resilin Autofluorescence by Age Class

As resilin fluorescence was not normally distributed (results of a Shapiro–Wilk test of normality: W = 0.817; *p* < 0.001), we used the ‘lm.rrpp’ function from the ‘RRPP’ R package to perform a permutational ANOVA, using age class, joint type, and an interaction term as explanatory variables and fluorescence as the response variable. We then performed a pairwise comparison test using the ‘pairwise’ function from the ‘RRPP’ R package to identify statistically significant differences among age classes at each wing joint. R code used in this analysis can be found at https://github.com/andrewckraemer/ResilinAnalysis_2023Apr18_ACK (accessed on 18 April 2023). 

## 3. Results

Resilin gene expression and autofluorescence were determined in multiple classes of workers and drone honey bees in this experiment. Significant differences in gene expression and autofluorescence were discovered among different groups examined, with higher rates of resilin expression and autofluorescence generally detected in the youngest age class examined.

### 3.1. Resilin Gene Expression

We found significant differences in resilin expression among the groups examined (F_4,161_ = 16.21; R^2^ = 0.29; *p* = 0.001). Specifically, the pupae age class expressed significantly more resilin than all other groups (*p* = 0.001, Figure 2, Appendix A).

### 3.2. Resilin Autofluorescence 

Resilin autofluorescence was best predicted by a model with the main effects of age class and joint type as well as an interaction term (Z = 15.21, *p* < 0.001). We also found significant differences in the expression of resilin among age classes depending on the joint type, except for all dorsal joints, both ventral and dorsal 2m-cu and Cu2-V joints, and 1m-cu on the ventral hindwing. Among joint types with significant differences, the newly emerged age class expressed the most resilin (Figure 3, Appendix A). The only other significant difference was significantly higher autofluorescence in the 1m-cu right forewing of the drone group compared to the forager group (*p* < 0.001). Representative images of each joint with significant differences in autofluorescence are shown in Figure 4. 

## 4. Discussion

This study determined how the expression and deposition of an elastomeric protein, resilin, varied with biological age class and caste in *A. mellifera*. Pro-resilin expression was high in worker pupae, and nearly absent in all post-emerging worker classes. Deposition of resilin is high in the ventral 1m-cu forewing and forewing and hindwing Cu-V joints of wings of newly emerged worker honey bees, but drops significantly in these joints post-emergence, although there was no further decline of resilin detected with worker age in this study. 

We confirm the location and presence of resilin at the 1m-cu, 2m-cu, Cu-V, and Cu2-V vein joints on the forewing and the Cu-V joint on the hindwing as described in *A. mellifera* by Ma et al. [27]. Additionally, we confirm the presence of a resilin stripe along the vannal vein of both the forewing and hindwing. Resilin also appeared to be present from the distal tip to along the costa yet was not analyzed because it could not be differentiated from potential edge effects.

The 1m-cu joint has been found to be of importance in previous studies on *Bombus impatiens*. Specifically, the presence of flexible resilin at the 1m-cu joint is important to flight stability, vertical force production, load-lifting capacity, and nearly 40% of the wing’s chordwise flexibility [15,17]. The flexibility of the 1m-cu joint also allows for the formation of a leading-edge vortex (LEV), a known important component of flight dynamics while also providing chordwise asymmetry (camber) to the wing to enhance lift generation [17,47,48,49]. Our study did not find a significant difference in resilin deposition levels at the 1m-cu joint among age classes, suggesting that the aforementioned flight mechanics do not change with age in *A. mellifera*.

There were no significant differences between the left and right wings at the group level in this study. Although the ventral sides of the wings had significant differences in resilin deposition among groups, there were no significant differences among groups for the dorsal sides of the wings in this study. This confirms the findings of Ma et al. [27], where *A. mellifera* resilin joints (patches) were typically located on the ventral side of the wing and were visually brighter during fluorescent microscopy. Studies on other species such as *Manduca sexta* also indicate differences between ventral and dorsal wing flexibility [13,14]. The significance of the ventral/dorsal resilin difference likely corresponds to the ventral flexion that occurs when the camber of the wing changes between concave and convex [50].

In this study, resilin deposition levels were maintained from the nest to the forager stage of workers in all joints studied. Historical data have indicated that post-ecdysis wings are highly sclerotized, dead cuticle [51]. However, this maintenance of resilin into older age suggests that some low level of hemodynamic support may be present in the wings. Specifically, the impacts of sclerotization of insect wings on hemolymph dynamics need to be investigated to determine if sclerotization prevents hemolymph flow and endocuticular deposition in adult *A. mellifera*. Research on the dragonfly *Zenithoptera lanei* generated the first evidence of a tracheal system in a flying wing for a modern adult insect [19]. Salcedo et al. [52,53] mapped the circulatory network of the wing of the grasshopper *Schistocerca americana* and confirmed the presence of hemolymph flow in all wing veins and determined the types of local flow patterns. These studies and others show that wing veins, containing hemolymph, trachea, and nerves, are not highly sclerotized in adult insect wings but allow for critical circulation.

The circulation of hemolymph may be key in maintaining the levels of resilin deposition. Resilin needs to be hydrated by water because it only exhibits elasticity and deformability while in aqueous media [29,54]. When resilin dries out, it becomes brittle and does not exhibit elastic properties. Beyond resilin, the wing cuticle requires hydration to maintain flexibility [20]. It is unknown if the living tissues within the wing veins continue to undergo endocuticular deposition. It is possible this could account for the consistent levels of resilin in wing vein joints in the oldest forager class in this study. Further investigation of hemolymph dynamics in adult *A. mellifera* wings may provide key insights into the biological functions of wings in adult insects and determine where resilin endocuticular deposition occurs and is maintained in adult insects.

## 5. Conclusions

The expression and deposition of resilin is dependent upon *A. mellifera* age class. This is the first study to demonstrate a relationship between resilin and *A. mellifera* biological age class and caste. The use of resilin as a marker of wing development, structural integrity, and potentially overall wing health should be investigated, especially in colonies affected by pathogens that affect wing development and morphology, such as deformed wing virus (DWV). DWV causes marked atrophy of wings during pupal development, resulting in honey bees that are unable to fly and are removed from the colony shortly after emerging [55]. DWV is a virulent virus that has been identified as a predictive marker of colony collapse disorder (CCD) [56]. Precocial foraging is another hallmark of sudden colony collapse in honey bees [37,57,58]. Precocial foraging results when the social structure of the colony collapses and young nurse or nest honey bees become foragers. This process is often driven by hormonal changes in the hive that are associated with stress and immunosuppression [7,57]. Future studies on the dynamics of resilin expression in normal versus diseased hives may help us to better understand the complex nature of honey bee disease, wing health, and colony loss mechanisms.

## Figures and Tables

**Figure 1 insects-14-00764-f001:**
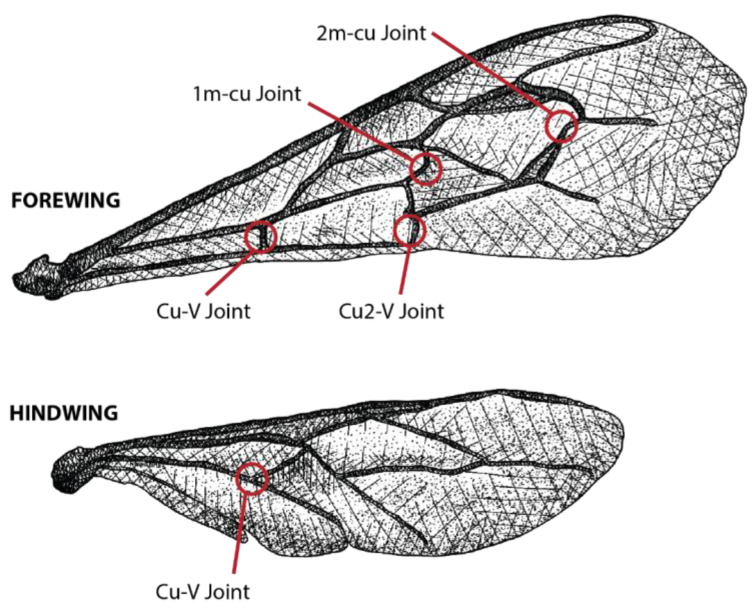
*Apis mellifera* wing joints analyzed in this study.

**Figure 2 insects-14-00764-f002:**
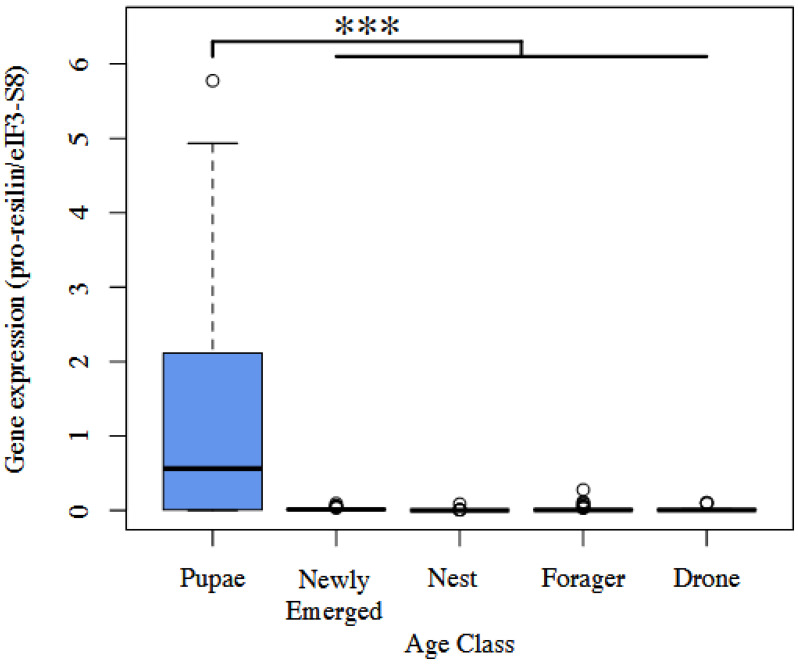
Box plot of pro-resilin expression normalized to eif3-S8 expression. *** = *p* < 0.001.

**Figure 3 insects-14-00764-f003:**
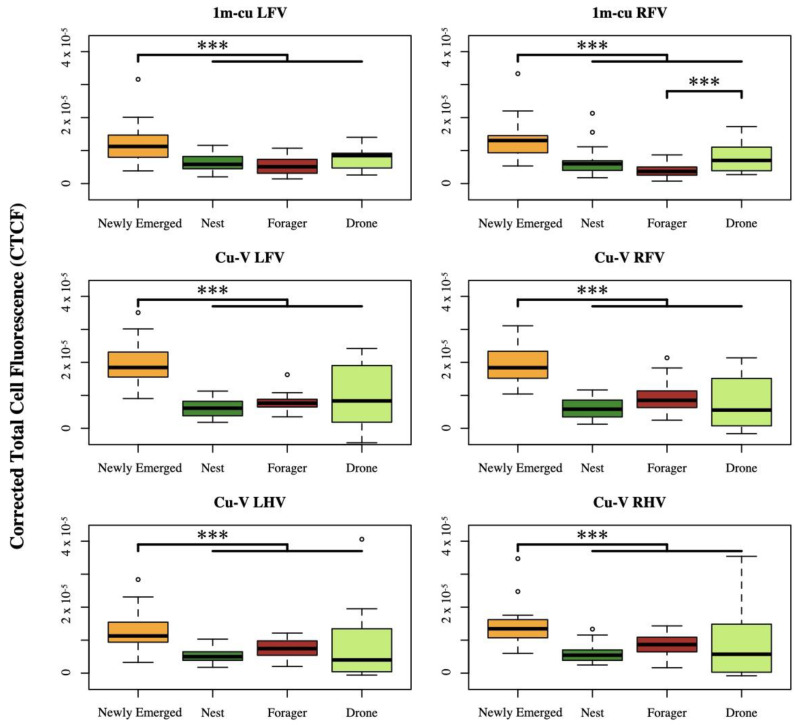
Box plot of mean corrected total cell autofluorescence (CTCF) by age class and caste according to wing joint. *** = *p* < 0.001. ° indicates an outlier.

**Figure 4 insects-14-00764-f004:**
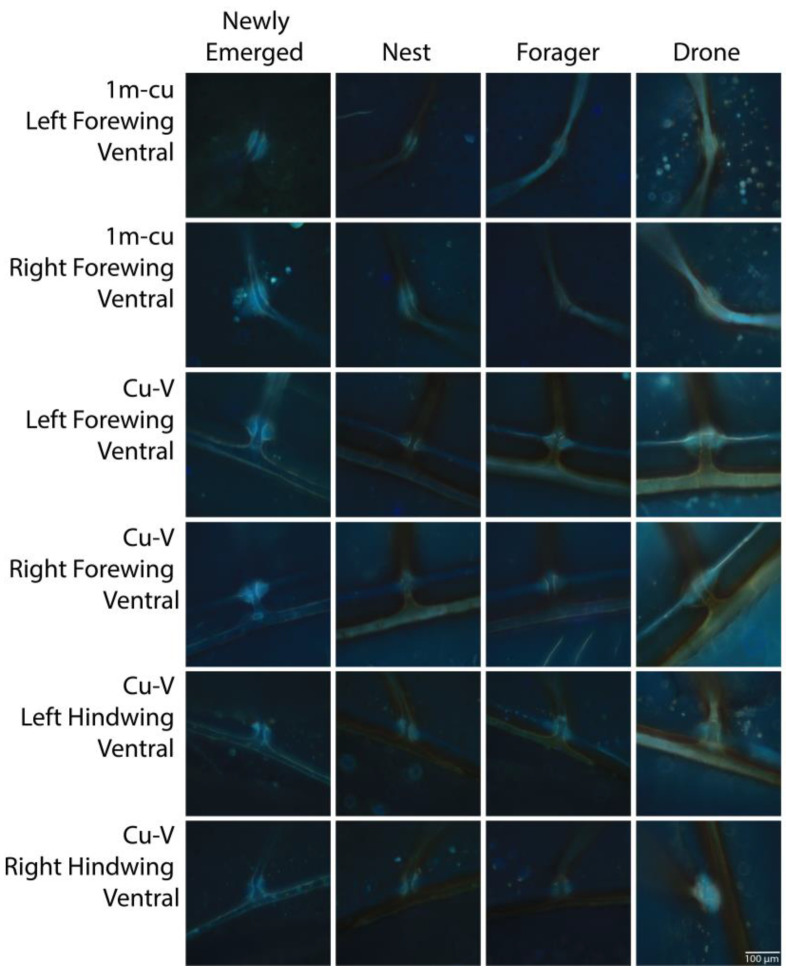
Representative images of 1m-cu and Cu-V joints on forewings and hindwings showing significant differences among groups (as shown in Figure 3). Measurement bar = 100 μm.

**Table 1 insects-14-00764-t001:** Primer and probe sequences.

Locus	Amplicon Size (bp)	Forward Primer	Reverse Primer	Probe	Accession Number
Pro-resilin	133	CGGTAATGGAGGTTATGG	CTCCATCTCTGCTTTCC	TCCAGATTGCTCGTCTTTCACTTC	XM_006563102.3
eIF3-S8	82	CGTACTGATCGTAGGGAA	CTACTGCTGGTCCAAGA	TTGCAATCCATAGCAGACACTCAT	XM_006564593.3

## Data Availability

Data are available through Dryad. DOI: 10.5061/dryad.wdbrv15tm.

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
