# Peer review of "Resilin Distribution and Abundance in Apis mellifera across Biological Age Classes and Castes"

_insects, 2023, doi:10.3390/insects14090764_

Round 1
Reviewer 1 Report
REVISION
Line 13: It cannot be assumed that resilin is only present in worker bees and not in drones or queens, so it is incorrect to write “worker bees called pupae”, because the young of queens and drones are also pupae.
Lines 11-14: The results reported in the abstract are contradictory.
Lines 75-76: Leave a space between the numerical value and the unit of measurement (323 nm…)
Line 117: Arrange spaces and dot: 420 nm [24]. The…
Line 137: second cubito-vannal
Line 155: honey bee
Line 313: Brittle
Line329: honey bees
Lines 246…, 285…, 322…: The sentences beginning respectively in lines 246, 285, and 322 are contradictory, as the generalisation of age dependence in the conclusions is not valid for all joints.
Line 285: is resilin deposit Perhaps you meant 'in'?
Lines 17-19, 285-287, 324: There is currently no reason to correlate resilin with the health status of honey bees, so no such claims can be made.
LEGENDA
In blue my suggestions, in red my corrections, highlight in yellow the parts to be removed.
COMMENT
Resilin distribution and abundance in different age classes of the honey bee workers and in drones have been examined.
The drafting of the manuscript is good, but discussion and conclusions are not entirely valid. It seems that the sentences beginning respectively in lines 246, 285, and 322, and in the first abstract are contradictory, as the generalisation of age dependence in the conclusions is not valid for all joints. I ask to the authors to better explain and not generalise the results. Furthermore, given the current lack of studies on the subject, it is risky to propose resilin as a marker of honey bee health.
The manuscript needs major revisions.
Reviewer 2 Report
This is an interesting study. It will be useful for others who study insect aging.
Lines 14 and 15 state that "...resilin... levels remain the same over time". Actually, the manuscript describes variation in resilin levels (lines 25-26, 234-235, Fig 2, Fig 3 etc).
The standard entomological term for the transition from pupa to adult is "eclose" or "eclosion". When an egg transitions to a larva, it "hatches". The manuscript should be corrected accordingly : lines 16 "after eclosing"; lines 28 and 42 "newly eclosed bees"; etc.
Some minor problems with grammar and syntax:
Lines 65-67 could be clarified. Perhaps they could read "The insect wings are composed of two layers of integument. Each of these layers contain cuticle, a chitinous, extracellular matrix. The wing veins contain six layers of cuticle. Inside the vein ..."
General grammar:
Line 36 "strategies allow"
Line 53 "rows of hamuli"
Line 81 "lines, the flapping"
Round 2
Reviewer 1 Report
II REVISION
Line 69: Remove a dot after cuticle
Line 78: Leave a space between the numerical value and the unit of measurement (420 nm)
Lines 93-94: Whati is the basis for this assumption? However, the potential of resilin to vary among A. mellifera biological age classes and be an age-dependent predictive marker of health has not been reported.
Lines 96-98: There is currently no reason to correlate resilin with the health status of honey bees, so no such claims can be made. And this hypothesis has not even been tested in the research! If resilin abundance and distribution significantly varies between healthy and diseased bees it could be established as a novel and predictive physiological marker of A. mellifera health.
Lines 109, 110, 115…honey bees
Line 311: Remove the dot after 52]
LEGENDA
In blue my suggestions, in red my corrections. Parts to be removed or modified are underlined in yellow.
COMMENT
The explicitly requested corrections have been made, but I expected that the relation between resilin and the honey bee's state of health would not be retained in the manuscript, and that the exact English terminology of Apis mellifera, honey bee, and not just 'bee', a term that identifies all bees, would be used everywhere! It is recommended that the authors correct all parts with these words or assertions.
I observed in the references several non-compliances with the journal rules regarding the Journal Name and the publication year.
Manuscript accepted with minor revisions.
